:Ọ: PLOS | ONE

# Control of visual adaptation depends upon task

**Mark Vergeer, Stephen A. Engel**[ID]*

Department of Psychology, University of Minnesota, Minneapolis, MN, United States of America

* engel@umn.edu

## Abstract

The visual system optimizes its functioning for a given environment through processes collectively called adaptation. It is currently unknown, however, whether adaptation is affected by the particular task the observer performs within that environment. Two experiments tested whether this is the case. Observers adapted to high contrast grating patterns, and the decay of adaptation was measured using a version of the tilt-aftereffect, while they performed two different secondary tasks. One task involved judging the luminance of a small circular spot at fixation, and was expected to be unaffected by adaptation. The other secondary task involved judging a low contrast grating, and adaptation was expected to make this task difficult by reducing the visibility of the grating. Identical displays containing both a fixation spot and a grating were used for both tasks. Tilt-aftereffects were smaller when subjects concurrently performed the grating task than when they performed the fixation task. These results suggest that the control of adaptation, in this case its decay, is sensitive to the nature of the task the observer is performing. Adaptation may attempt to optimize vision with respect to many different criteria simultaneously; task is likely one of the criteria included in this process.

**Data Availability Statement:** The data for the manuscript have been uploaded to figshare, and are available at this DOI: 10.6084/m9.figshare. 11833167

## Introduction

Visual exposure to an environment produces changes in visual function that improve neural efficiency and/or perceptual performance, via processes collectively known as visual adaptation. A paradigmatic case is dark adaptation, where exposure to low light levels engages a number of processes that allow us to see well in dim lighting.

It is not difficult, however, to find situations where adaptation actually *harms* performance on a given task. In the motion aftereffect, for example, prolonged viewing of a moving pattern may cause a physically stationary object to incorrectly appear as if it is moving (e.g., [1]). And in classical work on contrast adaptation, exposure to a high contrast grating can cause detection thresholds for similar gratings to more than double (e.g., [2]). Contrast adaptation can also cause differently oriented gratings to appear tilted several degrees away from their true orientation (the tilt aftereffect, e.g., [3]).

These negative effects arise in part because adaptation that is beneficial for some tasks can be detrimental for others [4]. For example, despite harming detection, contrast adaptation can aid performance on visual search tasks, by increasing the salience of targets that differ from the

**Funding:** SAE received BCS1558308 from the US National Science Foundation, who did not play a role in the study design.

**Competing interests:** The authors have declared that no competing interests exist.

adapter [5]. Thus, the net effects of adaptation on performance can be more or less beneficial, depending upon what tasks are being performed.

Is adaptation controlled in a way that considers these effects of task? If so, adaptation should depend on not just the environment to which the visual system is exposed, but also on what observers are doing within that environment. Here, we test this hypothesis, by examining whether task can affect the visual system's adaptive state. While short-term effects of task on basic visual processing have been reported [6], it is unknown whether it can affect the longer-term processes of visual adaptation.

In our experiments, observers adapted to a high contrast grating. Effects of contrast adaptation on orientation, known as tilt aftereffects (TAEs), were measured using a plaid pattern. Observers also performed one of two secondary tasks, on stimuli that were interleaved with the plaid. In one, observers were asked to judge low contrast gratings. In this task contrast adaptation would be expected to reduce the visibility of an already difficult to see stimulus, and so make performance difficult. In the other task, observers judged a relatively high contrast circle, and adaptation was expected to not affect performance. Importantly the displays presented during both secondary tasks were identical. We predicted that if adaptation depends upon task, then it should be reduced when observers perform the task where adaptation had the potential to hurt performance.

## Material and methods

### Participants

Ten volunteers (mean age: 21.2 years, SD: 1.8) participated in Experiment 1, and ten volunteers (mean age: 19.4 years, SD: 1.2) participated in Experiment 2. Our sample size was within the range of that used in the prior literature on contrast adaptation. All participants had normal or corrected to normal visual acuity, and gave written consent to participate under a protocol approved by the University of Minnesota IRB. The study was conducted in accordance with the Declaration of Helsinki. In Experiment 1, data from one participant was excluded from analysis, as they performed at chance level in the secondary task in multiple test sessions, indicating they were not following instructions.

### Apparatus

Stimulus presentation, timing and keyboard responses were controlled with custom software programmed in Python 2.7 using the PsychoPy library [7,8]. Head position was stabilized with a chin rest. In experiment 1, stimuli were generated by and presented on a 13" MacBook pro (1920x1080 at 60 Hz). Experiment 2 was controlled by a Mac Mini computer, that presented stimuli on a CRT screen (1024x768 pixels at 60 Hz). Mean display luminance was 42 candelas/meter$^2$; presented luminances were measured with a PhotoResearch PR-655 and the displayed levels were linearized using software look-up tables.

## Methods: Experiment 1

### Stimuli

The adapting stimulus was a full contrast vertical sinusoidal Gabor grating of size 10 x 10 visual degrees, with spatial frequency of 2 cycles per degree (cpd) of visual angle and envelope standard deviation (sd) of 1.66 deg (Fig 1A). During adaptation, the phase of the adapter was randomized at 10 Hz.

The test stimulus for the tilt aftereffect (Fig 1B) was a plaid made up of two 2 cpd sine wave component gratings symmetrically tilted from vertical and summed, which resembled a

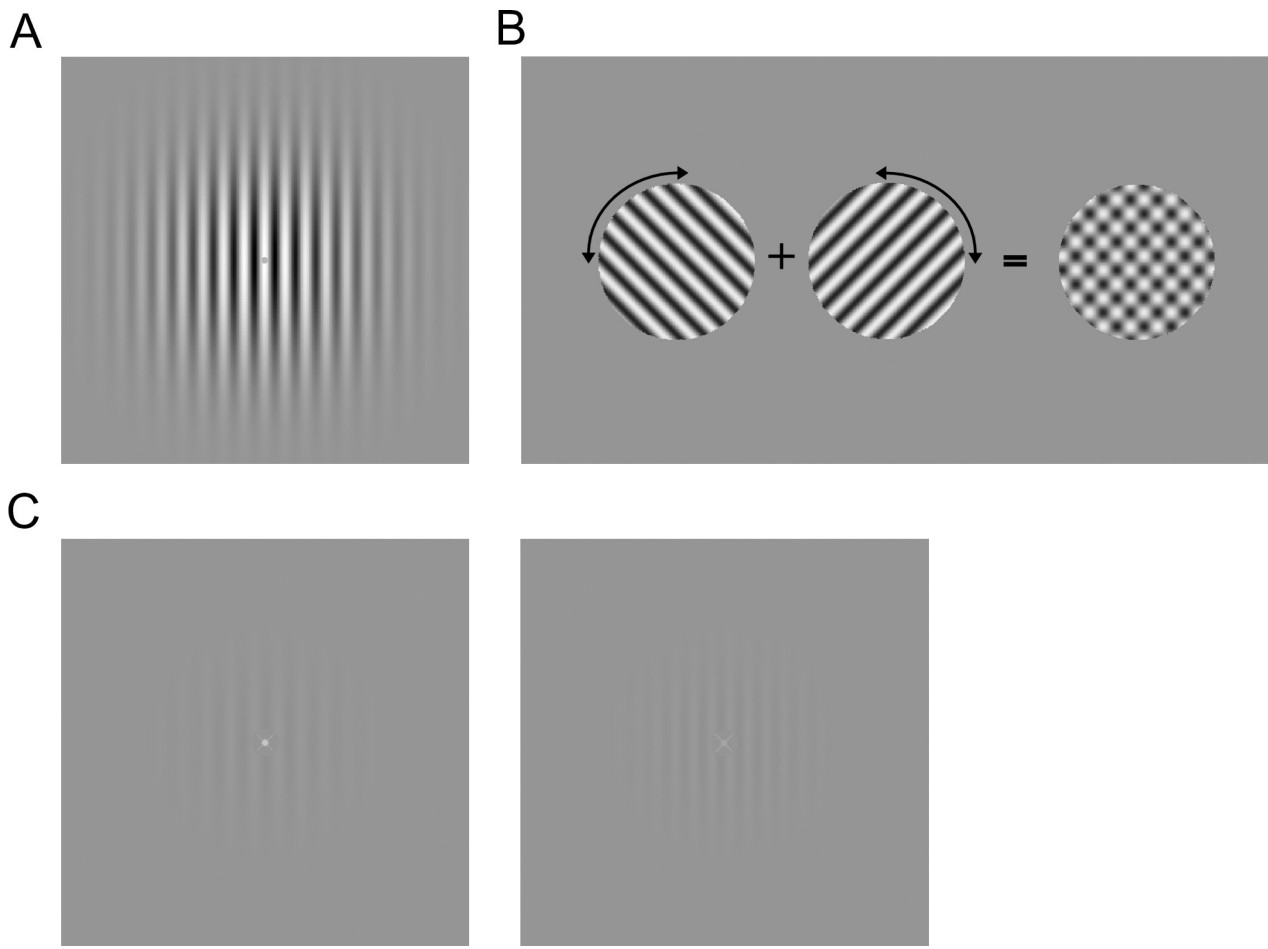

**Fig 1. Stimuli and tasks.** Panel A shows the adapting grating, a 2 cycle per degree Gabor pattern. Panel B shows the test stimulus used for the tilt aftereffect (TAE) task. Two orthogonal oblique sinusoidal gratings are combined to form a plaid of checks. In the TAE Task, participants adjusted the tilt of the oblique gratings, which changed the aspect ratio of the squares of the plaid, with the goal of making the checks appear square. Panel C shows the secondary test stimuli used for the secondary tasks. Two low contrast Gabors were displayed that differed in spatial frequency between the two images. A dot was displayed within a circular window at the center of each image, and the luminance of the dot differed between images. In the Grating task, participants judged which image had higher spatial frequency, and in the Fixation Task participants judged which dot had higher luminance.

blurred checkerboard pattern. The plaid subtended 4 deg of visual angle in a circular window and was presented at the fovea for 100 msec durations.

For secondary tasks, participants made discriminations performed on different aspects of the same secondary test stimulus: a low-contrast (2%) vertical 2 cpd Gabor grating with a 6 deg diameter and a spatial standard deviation of 1 deg, with a diagonally oriented fixation cross presented on a mean grey circle of 0.32 deg diameter in the center (see Fig 1C). A small dot of .06 deg diameter was superimposed on the center of the fixation cross. The secondary test stimulus was presented for 100 msec durations.

## Tasks

To quantify the strength of adaptation, we measured a version of the tilt-aftereffect (TAE) using plaid patterns [9]. We call this the *TAE Task*. When the component gratings of a plaid are tilted at +/- 45 degrees, the blurred checks that comprise the plaid are square, having identical height and width. With increased or decreased tilt (away from vertical) the checks become

rectangles that are wider or thinner than exact squares. Adapting to a high contrast vertical pattern generally causes a reduction in sensitivity to vertical, and a "repulsive" aftereffect where oriented patterns are perceived as tilted away from the adapter. In the present TAE task this is reflected by increased perceived tilt of the component gratings, making the checks look wider than they would look without adaptation.

The task for participants was to cancel these changes in check shape, to make the checks appear square, over a sequence of trials. In each trial, subjects viewed a plaid and adjusted the physical tilt of the gratings using 1 of 8 response keys (a, s, d, f, h, j, k, or l), that changed the orientation of the component gratings symmetrically by 2, 1.5, 1, 0.5, -0.5, -1, -1.5, and -2 degrees, respectively. The orientations of the gratings in the plaid were updated based on the subjects' response, and these updated orientations were presented in the next trial.

Participants also performed one of two secondary tasks, interleaved with the TAE task. Observers viewed versions of the secondary test display presented in two intervals. In one secondary task, called the *Grating Task* subjects performed a spatial frequency discrimination on the grating component of the display, indicating with a button press which of the two intervals contained a higher spatial frequency. In the other secondary task, called the *Fixation Task*, participants performed a luminance discrimination on the small dot in the display, indicating with a button press which interval was brighter.

In all trials, regardless of the task the participant was asked to perform, both spatial frequency and dot luminance differed in the two intervals, and the interval in which the higher spatial frequency grating was presented and the interval in which the higher luminance dot was presented were randomized independently. The size of the differences in spatial frequency and luminance were set at a level expected to produce 79% performance, as determined individually for each observer in a "pre-test" session, and were held constant throughout the main sessions.

## Design and procedure

The experiment consisted of a total of 6 sessions. In the first session participants trained on the TAE task, and the two secondary tasks, the Grating Task and the Fixation Task, each in separate blocks of trials. In the second session participants further trained the TAE task, and 79% correct thresholds were determined for both secondary tasks. Thresholds were measured in 3 blocks of 50 trials each. For each task, the stimulus difference used in the main experiment was set to the median threshold measured in the pre-test. In addition, participants performed a TAE pre-test in which 3 minutes of adaptation was followed by 1 minute of the TAE task. This was used to determine the initial orientations of the component gratings for the TAE task in the main sessions for the TAE task; main session blocks began with a test stimulus set to half the maximum TAE obtained in the pre-test.

In the four main sessions, participants performed one of the secondary tasks followed by TAE task (Fig 2). In these sessions, secondary task trials were immediately followed by TAE trials. Each trial lasted 1.5 sec; in secondary task trials two 100 msec intervals were followed by a 1.3 sec response interval, while for TAE trials, one 100 msec stimulus presentation interval was followed by a 1.4 sec response interval.

The order of blocks in the main sessions is shown in Fig 3. Each main session started with a 2 min block of practice trials interleaving the TAE task with the secondary task for that particular session. Next, pre-adaptation baseline performance was measured, again in a 2 min block of trials. This was followed by 3 minutes of adaptation to vertical, and a 2 min block of trials to measure post adaptation performance. In two sessions the secondary task was the Grating Task, and in the other 2 sessions it was the Fixation Task, with order counterbalanced between participants.

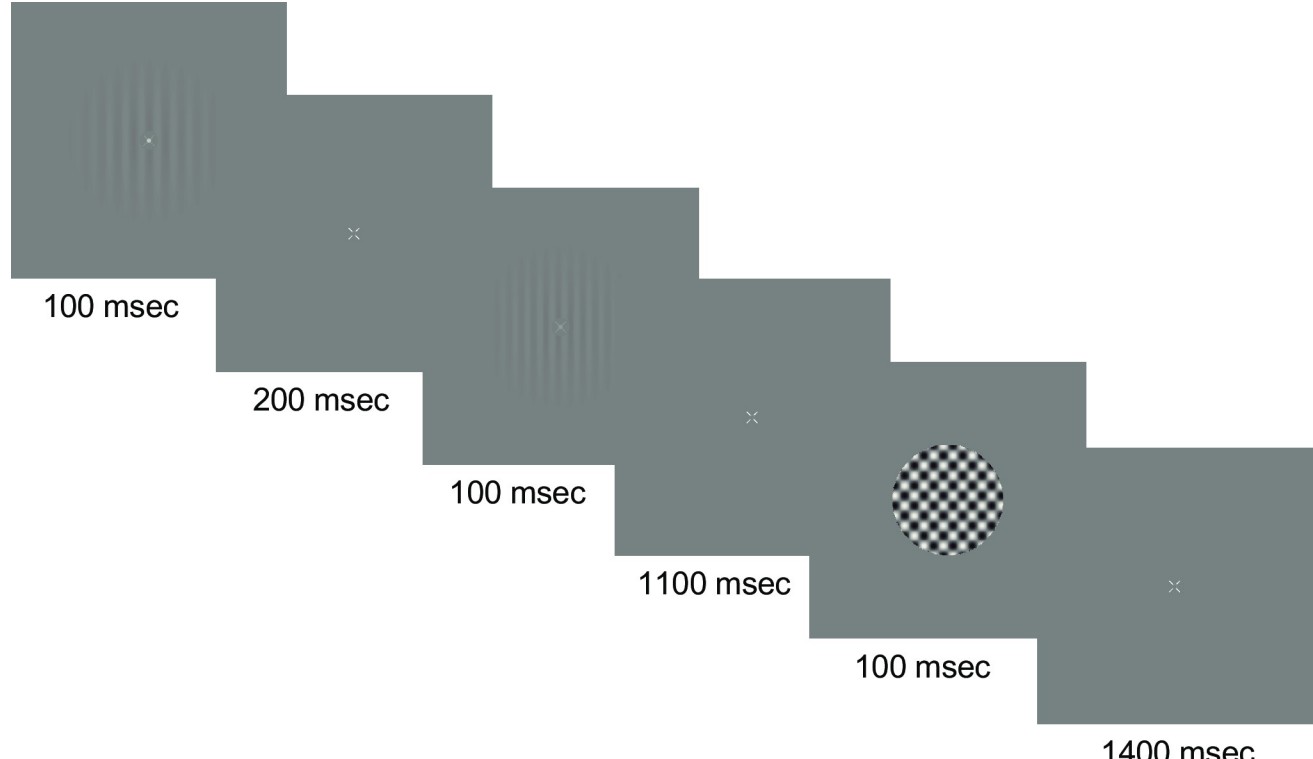

**Fig 2. Stimulus sequence.** In the main testing sessions, a secondary task trial was immediately followed by a TAE Task trial. For the secondary task, the two stimuli were each displayed for 100 msec with a 200 msec gap between and an 1100 msec response period. For the TAE Task a 100 msec test presentation was followed by a 1400 msec response period.

## Data analysis

TAE results were plotted as estimates of the orientation of the plaid components that appeared square. These were computed as the average presented orientation on each trial. Because the presented orientation was determined by a staircase-like procedure (based on the participants' responses that attempted to make the checks appear square; see Tasks above), it is possible that simply averaging the presented orientations would produce a biased estimate. To test whether simple averaging of the staircase levels was a reasonable estimate, we performed a Monte-Carlo simulation of our experiment with a model observer responding. The model observer's orientation that appeared square started at 48 degrees and decayed exponentially over time with a time constant 0.03, which produced a "true" effect similar to those seen in our and other studies. On each simulated trial, noise was added to the presented orientation, and the observer used a simple multiple threshold decision model to pick its response. Averaging across 10,000 simulations, the estimate obtained from simple averaging of the presented orientation fell close to the model observer's true effect and did not reliably differ from it.

## Methods: Experiment 2

### Stimuli and tasks

Stimuli were identical to those used in Experiment 1, except for the secondary task display, where in one of the two images the grating was oriented horizontally and in one it was oriented vertically. The interval in which a particular orientation was presented alternated from trial to trial.

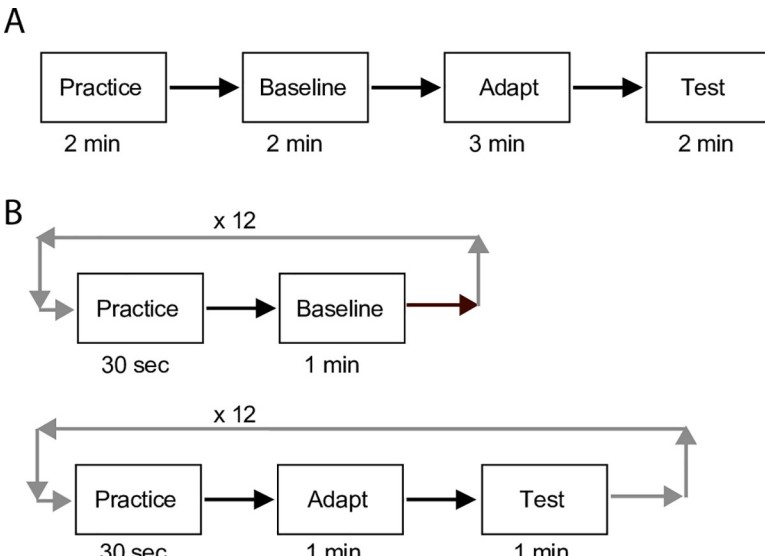

**Fig 3. Design of experiments 1 and 2.** Panel A shows the sequence and duration of blocks in the main sessions of Experiment 1. A 2 min block of practice was followed by a 2 min block of trials prior to adaptation (Baseline). Then 3 minutes of viewing the adapter stimulus was followed by an additional 2 min of trials (Test). Panel B shows the sequence of blocks for Experiment 2. Prior to adaptation, 30 sec practice blocks alternated with 1 min Baseline blocks. Twelve blocks of each were performed with the secondary task switching after each Baseline block. Next, during adaptation, 30 sec practice blocks alternated with 1 min blocks of trials (Test) with 1 min of the adapting grating presented in between. Again, twelve blocks of each were performed, with the secondary task switching after each Test block. (see text for details).

## Design and procedure

In an initial session, participants trained on the TAE, Grating, and Fixation tasks, following a similar procedure as in Experiment 1. In the second session participants further trained the TAE task and 79% correct thresholds were determined for the secondary tasks. Both secondary tasks were performed in 3 blocks of 50 trials each. For each task, the stimulus increment used in the main experiment was set to the median threshold measured in the pre-test for each participant. In addition, participants performed a pre-test in which 1 minutes of adaptation was followed by 1 minute of the TAE task. The maximum TAE from this pre-test was used as the starting value of the TAE task in adaptation blocks of the main sessions. The average of the median tilt values from the final two TAE task practice blocks (without adaptation) was used as the starting value of the TAE task in main session baseline blocks.

The two main sessions contained more blocks of trials than in Experiment 1, and both secondary tasks were performed in separate blocks in each session. Each main session started with 12 baseline blocks of 1 minute each (see Fig 2). In each block the TAE task was followed by either the Grating Task or the Fixation Task. Which secondary task was performed alternated between blocks, and participants started with a different secondary task in each of the 2 sessions. Note that secondary task display was identical, regardless of which secondary task was performed. To aid transition between secondary tasks, each block was preceded by 30 seconds of practice on the task combination that had to be done during that particular block. Trials were as in the previous experiment.

Following the 12 baseline blocks, participants performed 12 adaptation blocks. These were identical to the baseline blocks, except that the secondary task practice was followed by 1 minute of adaptation to a full contrast vertical grating.

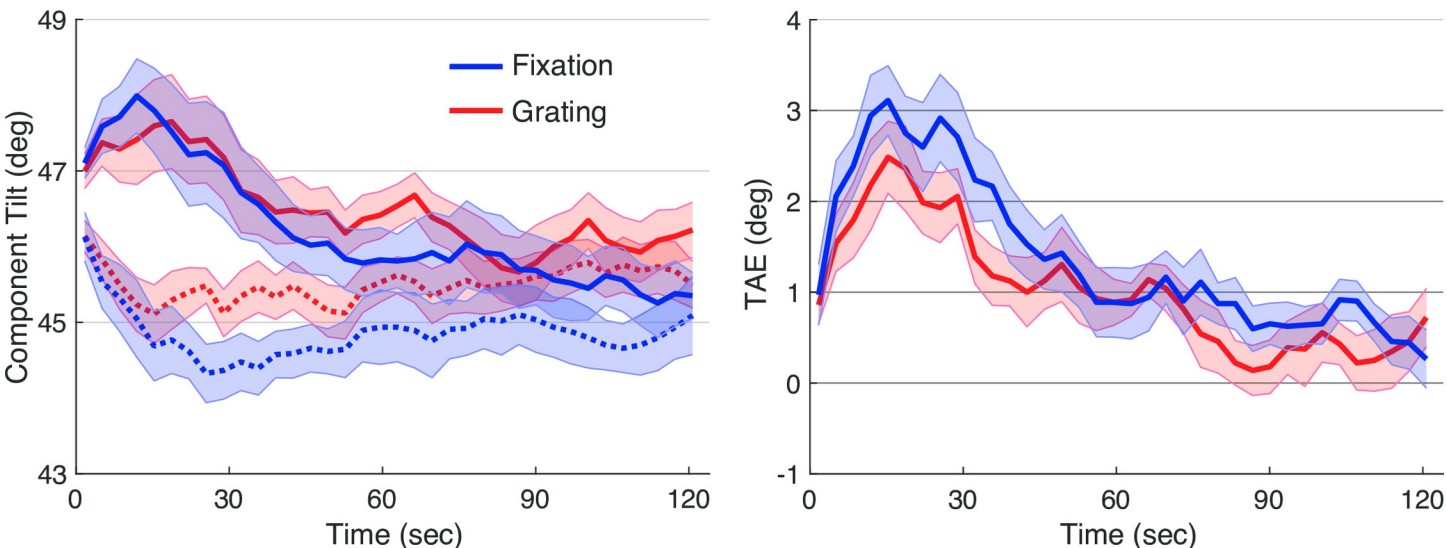

**Fig 4. Results of experiment 1.** The left panel shows results of the TAE Task, when interleaved with the Grating Task (red) and with the Fixation Task (blue), during baseline (before adaptation., dotted) and following adaptation (solid). The right panel shows effects of adaptation computed by subtracting the baseline TAE from TAE after adaptation, for each secondary task, separately. Lines plot means across participants and error ribbons indicate +/- one standard error of the mean.

## Results: Experiment 1

The goal of Experiment 1 was to test whether adaptation, as measured by the TAE, was influenced by performance of an additional task. Our hypothesis was that participants would show more rapid decay of adaptation when it would be expected to hinder task performance. The secondary task was either a spatial frequency discrimination task (the Grating Task) or a brightness discrimination task on a small dot presented at the fovea (the Fixation Task). Adaptation should be costly when performing the Grating Task, since it reduces sensitivity to vertical, making the test grating difficult to see, and presumably to judge, potentially lowering performance. Conversely, the vertical gratings were distractors while performing the Fixation Task, and reducing sensitivity to them should, if anything, aid performance.

Effects of adaptation, measured with the TAE task, are shown in Fig 4. Plotted points are estimates of the orientation of the plaid components that appeared square, computed as the average presented orientation on each trial (Fig 4, left). Baseline trials hovered around 45 degrees (dotted lines; after the adaptive procedure moved from its starting value over the first few trials); this orientation that produces physically square checks. Three minutes of adaptation to the vertical grating produced a repulsive aftereffect, and observers cancelled this effect by setting the component gratings closer to vertical (solid lines). This adaptation decayed over time. To account for across subject differences in baseline (the plaid configuration that appeared square without adaptation), we computed a net TAE by subtracting baseline from adaptation for each observer. This net TAE score is plotted in Fig 4, right.

The data showed an overall trend towards more rapid decay of adaptation in the Grating condition than in the Fixation condition. We computed the total tilt-aftereffect by taking area underneath the baseline-corrected TAE time courses. This score was reliably larger in the Fixation condition, as tested with a paired samples t-test ($t(8) = 2.3$, $p < 0.05$).

As an exploratory analysis, we examined the TAE separately for the first and second sessions in which subjects performed each task (Fig 5). We reasoned it was possible that learning to perform the concurrent tasks could affect our results. In the first session, there was again a

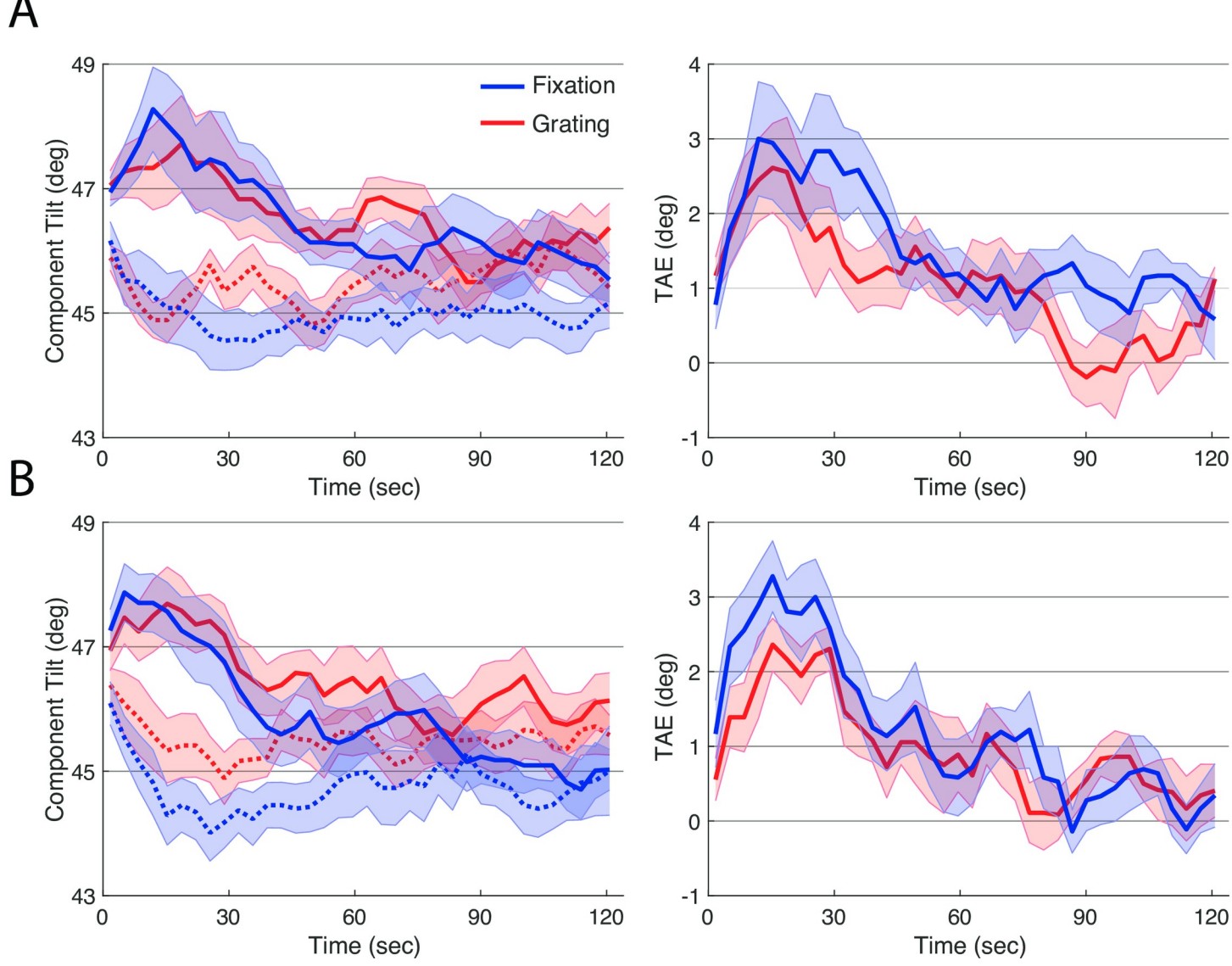

**Fig 5. Results of experiment 1 by session.** The left and right panels show raw results and effects with baseline subtracted, as in Fig 5. Panel A shows results for the first session, and panel B for the second session. Plotting conventions are as in Fig 4.

reliably larger TAE in the Fixation task as compared to the Grating task ($t(8) = 2.3$, $p < 0.05$). This difference was smaller and less reliable in the second session ($t(8) = 1.5$, $p < 0.17$).

Finally, we examined whether our assumption regarding task—that performance on the Grating task would be more challenged by adaptation than performance on the fixation task—was reflected in the data. Because adaptation was strongest in the first 40 sec of testing, we computed average performance over that interval and the two successive 40 sec periods. Performance on the Grating task was reliably reduced following adaptation during this first interval (Fig 6; $t(8) = 2.8$, $p < 0.03$). Performance was reduced numerically, but not reduced reliably for the Fixation task ($t(8) = 1.3$ $p > 0.2$), and the difference between the two tasks was not reliable ($p > 0.5$).

Overall, results generally supported the hypothesis that adaptation is affected by task: Total TAE was smaller when observer performed the Grating Task, than when they performed the

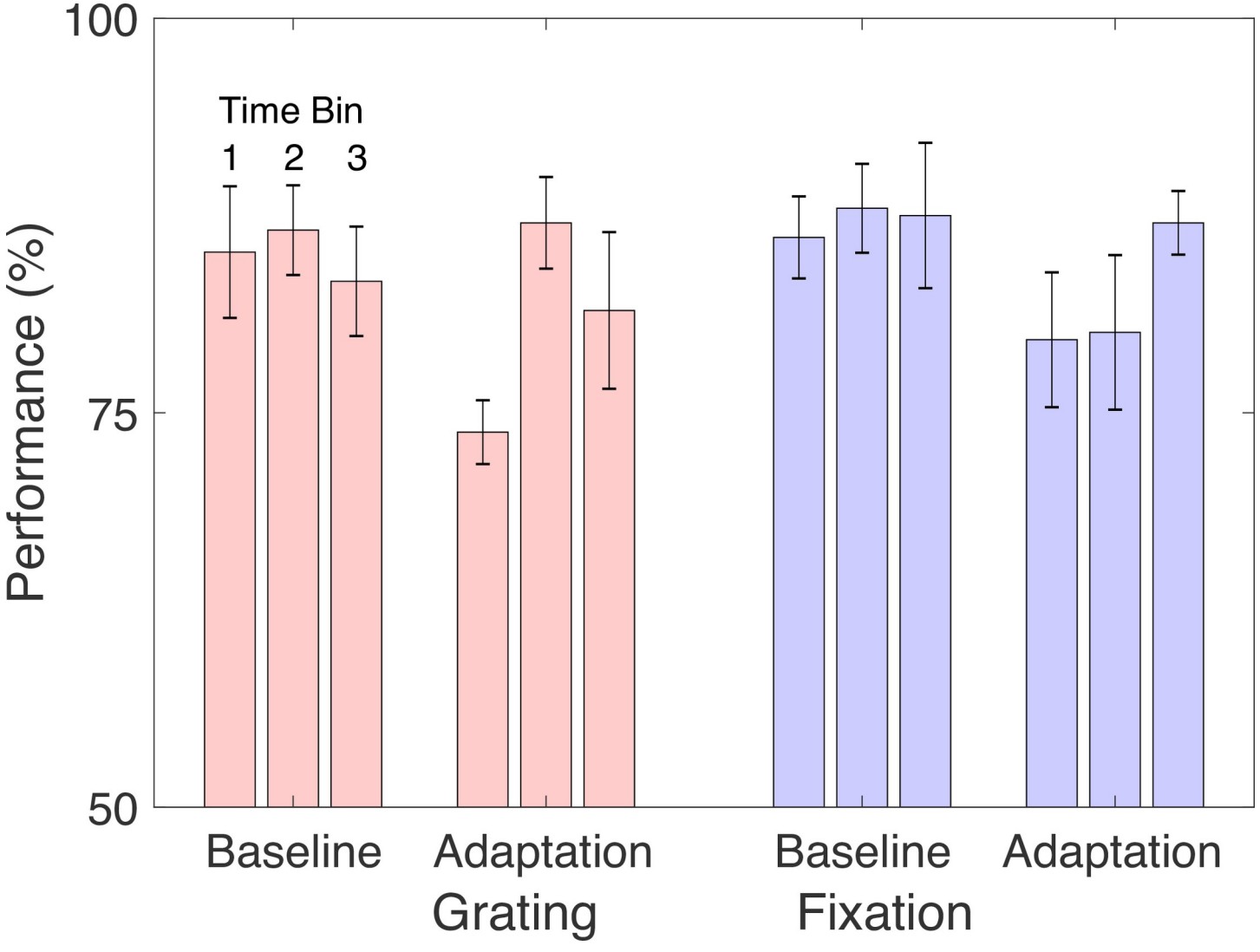

**Fig 6. Performance on secondary tasks.** Mean performance on the Grating Task and the Fixation Task shown during baseline (before adaptation) and following adaptation. Bars represent three 40-sec time bins over which performance was computed. Error bars represent +/- one SEM.

Fixation Task. Additionally, supporting our assumption when designing the tasks, performance in the Grating Task was affected by adaptation, while for the Fixation Task it was not. We hypothesize that because adaptation lessened the effective visual contrast available for performance of the Grating Task, the visual system counteracted its effects, causing adaptation to decay more rapidly.

One complication in the data is that performance on the plaid task differed as a function of task during the baseline blocks. Differential effects of attention in the two conditions may explain these differences. When participants attend the gratings in the secondary task display, they may have produced the small amount of contrast adaptation that was measured by the TAE task during baseline. This effect may have been reduced in the Fixation Task, when subjects did not attend the gratings. Such an effect, however, should also arise during the adaptation blocks, and in this case subtracting the two would cancel it out.

It remains possible, however, that the baseline differences were due to unrelated factors, such as noise of some sort, in which case they may explain, all by themselves, the differences in

total TAE observed between tasks. Accordingly, we designed Experiment 2 to minimize possible differences in baseline performance.

## Results: Experiment 2

Experiment 2 modified procedures to reduce baseline differences, and added other improvements to the design. First, to minimize possible contrast adaptation during the secondary task at baseline, we changed the orientation of one of the two grating presentations in the secondary task display. Specifically, within each trial we presented one of the gratings horizontally and one vertically, which should reduce the orientation-specific adaptation that is measured by the TAE task. This change was made both during baseline and adaptation blocks.

We also changed the starting values of the component gratings in the TAE task. In Experiment 1, the orientation of the gratings was initialized to the same value in both baseline and adaptation blocks. This caused us to underestimate the magnitude of the TAE at the beginning of the adaptation blocks, where it was expected to be strongest. In Experiment 2, the test plaid was initialized in adaptation blocks to match the observer's peak TAE level, as estimated in a pre-test, while baseline blocks started at the participant's baseline TAE level, also estimated during a pre-test.

Finally, to increase the reliability of our data in Experiment 2, both secondary task conditions were run in each session, and each was repeated 6 times per session, in alternating 1 min blocks. In order to accomplish this in a reasonable total session length, we shortened the adaptation duration from 3 minutes to 1 minute (See Fig 3B, Methods section).

As in Experiment 1 we expected adaptation to decay faster for the task where it would be detrimental to performance, i.e. the Grating Task. Fig 7 show the results for Experiment 2, and there was an overall trend in this direction. Note that the curves are much smoother because they are averages of many more blocks per condition. Total adaptation, as measured by area under the TAE decay curve, was numerically weaker during performance of the Grating Task than the Fixation Task, but this difference was not reliable ($t(8) = 1.8$, $p < .16$). Because exploratory analysis of Exp 1 revealed stronger effects of task in the first session, we next examined results for each session separately.

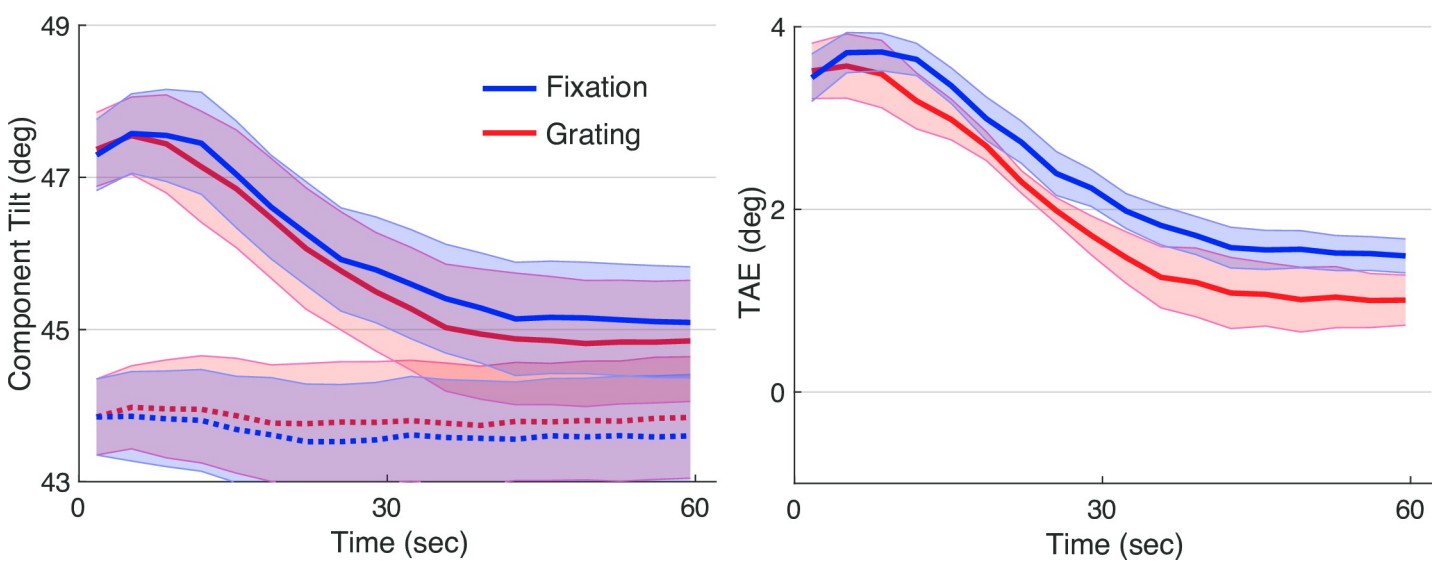

**Fig 7. Results of experiment 2.** The left panel shows results of the TAE Task, when interleaved with the Grating Task (red) and with the Fixation Task (blue), before (dotted) and after adaptation (solid). The right panel shows effects of adaptation computed by subtracting baseline TAE from TAE after adaptation, for each secondary task, separately. Lines plot means across participants and error ribbons indicate +/- one standard error of the mean.

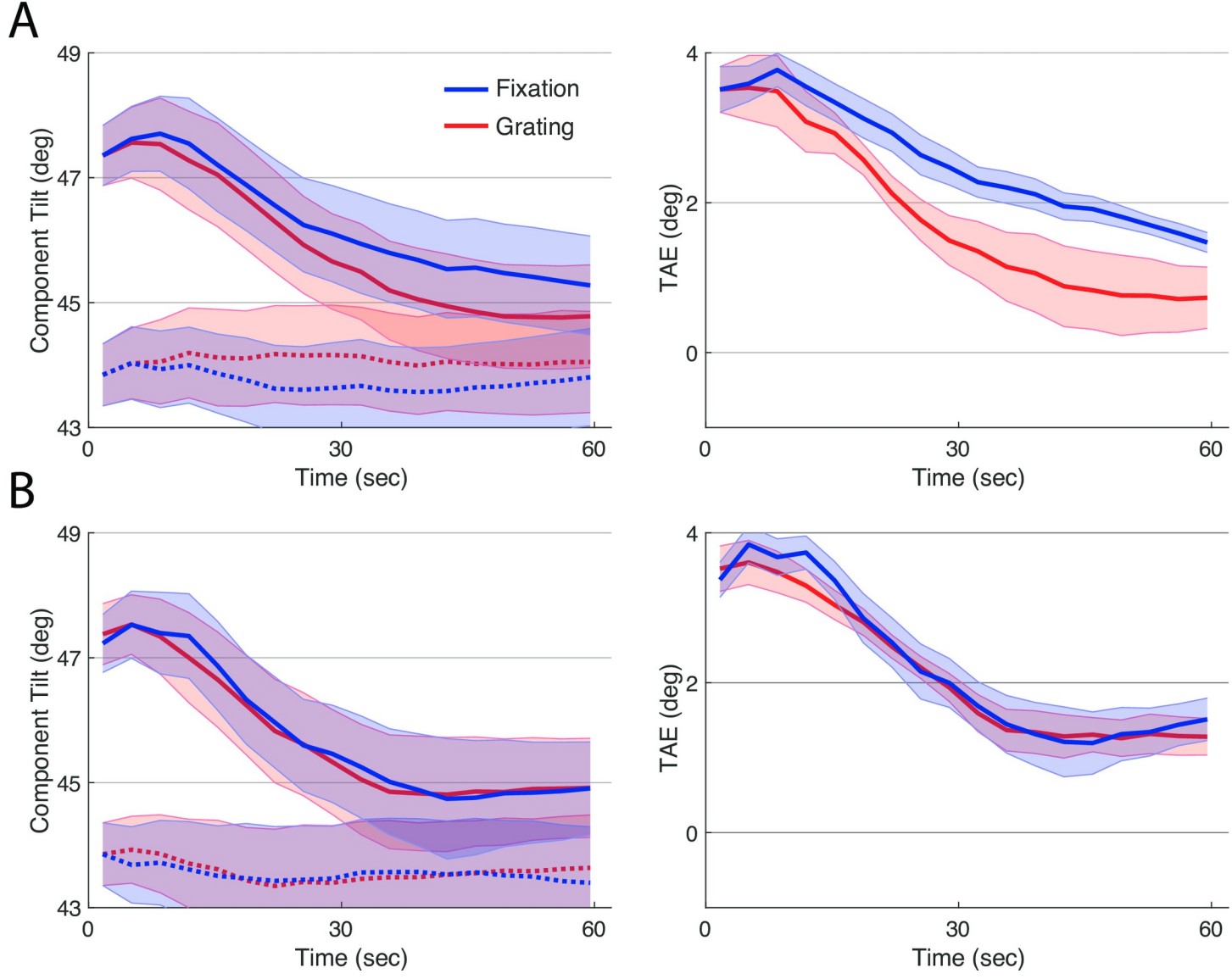

**Fig 8. Results of experiment 2 divided by session.** The top and bottom panels show results for the first and second session, respectively. Plotting conventions are as in Fig 7.

In the first session, total adaptation was reliably weaker when subjects performed the Grating task than when they performed the Fixation task (Fig 8A; $p < 0.01$, non-parametric signed-rank test used because of non-normal distribution of data). The difference was not reliable in the second testing session (Fig 8B; $p > 0.5$). In addition, while there was a small baseline difference between tasks in the first testing session, the difference in total adaptation was reliable even without correcting for baseline ($p < 0.02$, non-parametric signed-rank test).

We again analyzed the performance on the secondary tasks, before and after adaptation (Fig 9). Performance on the Grating task was again reliably reduced following adaptation during the first interval ($t(8) = 3.9$, $p < 0.01$). Performance was reduced numerically, but not reliably for the Fixation task ($p > 0.5$). The difference between the two tasks was not reliable ($t(8) = 1.9$, $p < 0.09$).

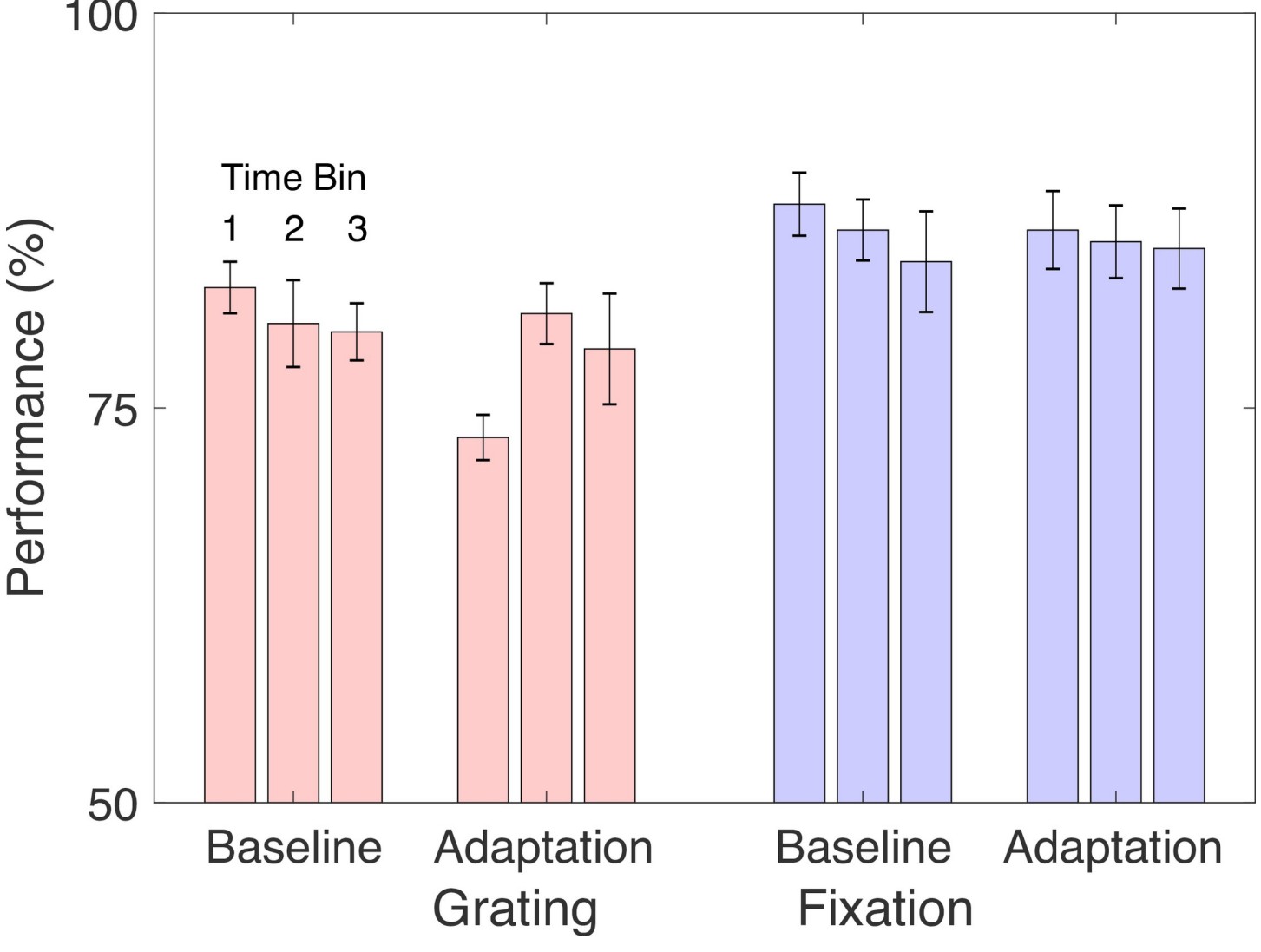

**Fig 9. Performance on secondary tasks.** Mean performance on the Grating Task and the Fixation Task in Exp. 2 is shown. Plotting conventions are as in Fig 6.

A final exploratory analysis examined why the differential effects of task were stronger in the first session in both experiments. One possibility is that with practice, people learn to switch between different adaptive states for the TAE and secondary tasks, leading to greater independence of the two. Past work has suggested that learning can influence adaptation [10,11,12]. Examining secondary task performance from both experiments, separated by session, supports this explanation (Fig 10). As expected, adaptation had the largest effect on the Grating Task. However, this mainly occurred during the first testing session (where the Grating Task showed a significantly larger decrement in performance than the Fixation Task at the first time bin ($t(17) = 2.5$ $p < 0.025$)), consistent with the possibility that the secondary task became more independent of adaptation over time.

## General discussion

Two experiments measured the strength of adaptation while subjects performed two concurrent tasks: one in which adaptation was expected to be detrimental to performance and one

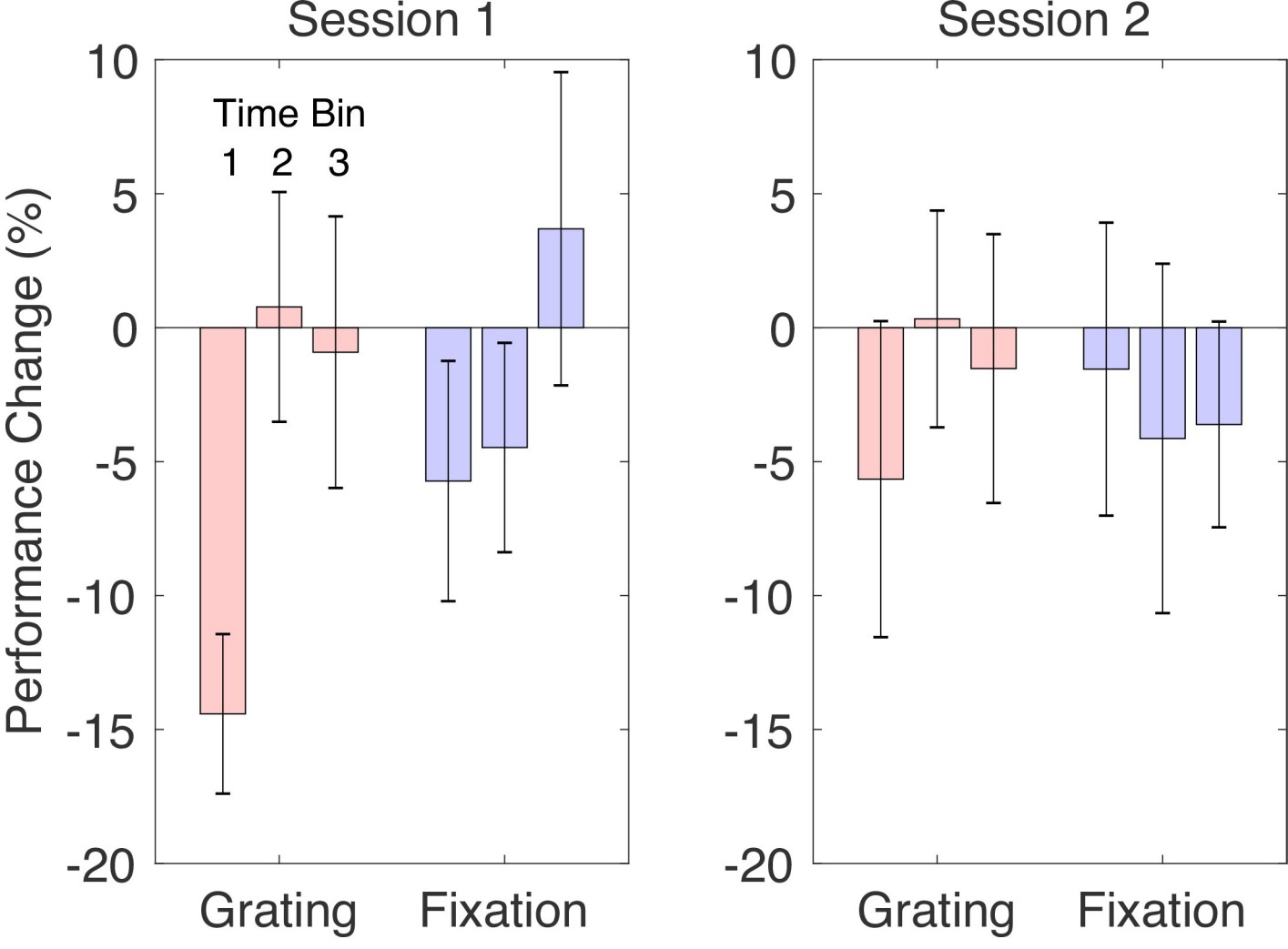

**Fig 10. Pooled secondary task performance.** Mean change in performance between baseline and adaptation conditions for the Grating Task and the Fixation Task is shown, pooled across both experiments. Other plotting conventions are as in Fig 9.

where it had little impact. More rapid decay of adaptation was observed in the case where it was expected to be detrimental to performance. These results suggest that the processes controlling adaptation are sensitive to the task the observer is performing.

Adaptation is generally theorized to be a way in which the visual system optimizes its function (for reviews, see [13,14,15]). Most of these theories propose that adaptation improves neural representations of stimuli, without considering the particular task the observer is currently performing but see [4]. Our results argue that such theories are incomplete, and that visual task performance must be taken into account in the control of adaptation.

Prior work has revealed other ways in which the visual system adjusts to aid performance of the task. For example, performing tasks that require high visual acuity appears to increase response in neurons that represent the high spatial frequency information required for the task [6]. Relatedly, visual attention can enhance neural responses that are most informative for a given task [16]. In our experiments, task performance appeared to alter not just immediate neural responsiveness, but also the longer-term effects of visual adaptation.

More generally, because the visual display was identical in both task conditions, our results also indicate that adaptation is not determined by the stimulus alone. This interpretation agrees with past work showing that changing visual attention can influence the amount of adaptation a given stimulus produces (e.g. [17,18]). Whether attention can affect adaptation, and by how much, remains a topic of some debate [18,19]. Both sides of this debate assume a similar "standard" model of how attention and adaptation interact: Attention strengthens certain neural responses to the attended stimuli in early visual cortex, and these larger responses in turn produce greater amounts of adaptation.

This standard model cannot explain our results. The adapting stimulus, a high contrast vertical grating patch, was presented under identical neutral attention conditions in all conditions (and prior to task performance). Our experiment measured effects of the secondary task on the decay of adaptation produced under this common condition. It is theoretically possible, however, that attention influenced some amount of adaptation arising from the secondary task display itself. But such effects are not likely to play a role in our results, since they would be expected to go in the opposite direction from what we observed. In the Grating Task, attention to the grating should boost the neural response it produces, which would be expected to produce adaptation that decreases response to subsequent low-contrast stimuli. Such an effect would be in the same direction as that produced by the high contrast adapting grating, and so should enhance the overall amount of adaptation seen. Importantly, the effect we observed was opposite to this—the Grating Task reduced overall adaptation.

Attention is, however, a potential mechanism through which task could affect adaptation. In the Grating Task, for example, attention to the low contrast grating could increase the amount of adaptation to that grating. Adaptation to low contrast generally produces the opposite effect of adaptation to high contrast (e.g. [20]), and so would raise the gain of neurons whose gain responsiveness was reduced by the initial adaptation, causing more rapid decay of the TAE.

Note that this account proposes a gating effect of attention, on adaptation, which differs from the standard model discussed above. Functionally, this attention-mediated, task-based adaptation to low contrast could serve to center the response to the grating on the steep part of the responding neurons' contrast response curve (e.g. [21]). Alternatively, the same adaptation might be characterized as increasing the signal-to-noise of the most important information for the task (e.g. [22]). These accounts are not mutually exclusive.

The orientation specific contrast adaptation measured by the TAE task in the present work likely originates to a large extent in primary visual cortex (V1), the first in the hierarchical stream of visual areas in cortex. It is possible that effects of task could influence processes controlling adaptation within this early stage. Adaptation is inherited by later stages of visual processing [23,24], however, and so task could also affect responses there.

Our results are limited in several ways, however. First, the empirical results supporting our assumption that adaptation harms task performance more for the Grating Task than for the Fixation Task could be stronger. While we indeed find an effect of adaptation on performance for the Grating Task, and find no effect on the Fixation Task, the interaction between adaptation and task was only statistically significant when pooling over both experiments. Second, and more critically, the effect of task on adaptation was seen primarily during the first testing session. Our account of why this may be so- that participants learned to perform the secondary task independently of the effects of adaptation- is speculative, though plausible based on past work [11]. It additionally suggests that in some circumstances adaptation may not need to be regulated by task.

Finally, relative to overall effects of adaptation, its modulation by task was not large in absolute terms. Characterizing when task performance can affect adaptation, and by how much,

including in natural viewing during natural tasks, is an important line for future research. Nevertheless, our results provide reasonably strong support that task can affect the strength of adaptation.

## Conclusions

Visual adaptation likely optimizes the visual system with respect to many different criteria simultaneously. Criteria identified in past work include the precision with which individual neurons can represent changes in the visual stimulus (e.g., [25]), the representational capacity of a collection of neurons (e.g., [26]), the independence with which different neurons produce spikes (e.g., [27]), the ability to maintain perceptual constancy (e.g., [28]), the ability of neurons to respond robustly to novel patterns of stimulation [29], and others (many reviewed by [13,14,15]). Many of these criteria may be closely related, and could be subserved by common neural mechanisms, while others may conflict with one another. Regardless, the present results add task performance to this list. How the visual system optimizes for many criteria simultaneously remains an important question for future research, both theoretical and empirical.

## Acknowledgments

We thank Allyson Bigelow for assistance in testing participants.

## Author Contributions

**Conceptualization:** Mark Vergeer, Stephen A. Engel.

**Data curation:** Mark Vergeer, Stephen A. Engel.

**Formal analysis:** Mark Vergeer, Stephen A. Engel.

**Funding acquisition:** Stephen A. Engel.

**Investigation:** Mark Vergeer, Stephen A. Engel.

**Methodology:** Mark Vergeer, Stephen A. Engel.

**Project administration:** Stephen A. Engel.

**Resources:** Stephen A. Engel.

**Software:** Mark Vergeer, Stephen A. Engel.

**Supervision:** Stephen A. Engel.

**Validation:** Mark Vergeer.

**Visualization:** Stephen A. Engel.

**Writing – original draft:** Mark Vergeer, Stephen A. Engel.

**Writing – review & editing:** Mark Vergeer, Stephen A. Engel.

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
