## [Decision Letter · Decision Letter 0]

19 Nov 2019

PONE-D-19-23461

Adaptation is weaker when it harms task performance

PLOS ONE

Dear Dr. Engel,

Thank you for submitting your manuscript to PLOS ONE. After careful consideration, we feel that it has merit but does not fully meet PLOS ONE’s publication criteria as it currently stands. Therefore, we invite you to submit a revised version of the manuscript that addresses the points raised during the review process.

Both reviewers have raised significant issues concerning the stimuli including gamma-correction of the display, threshold estimation procedure and potential confounding effects of differences in baseline between the "fixation" and "grating" conditions. There are also major concerns about the quality of the figures (see below). Please address each of these issues in your revised manuscript and rebuttal. 

We would appreciate receiving your revised manuscript by Jan 03 2020 11:59PM. To enhance the reproducibility of your results, we recommend that if applicable you deposit your laboratory protocols in protocols.io, where a protocol can be assigned its own identifier (DOI) such that it can be cited independently in the future. For instructions see: http://journals.plos.org/plosone/s/submission-guidelines#loc-laboratory-protocols

We look forward to receiving your revised manuscript.

Kind regards,

Mark W. Greenlee

Academic Editor

PLOS ONE

Journal Requirements:

Reviewers' comments:

Reviewer's Responses to Questions

**Comments to the Author**

1. Is the manuscript technically sound, and do the data support the conclusions?

Reviewer #1: No

Reviewer #2: Yes

2. Has the statistical analysis been performed appropriately and rigorously? 

Reviewer #1: Yes

Reviewer #2: Yes

3. Have the authors made all data underlying the findings in their manuscript fully available?

Reviewer #1: Yes

Reviewer #2: Yes

4. Is the manuscript presented in an intelligible fashion and written in standard English?

Reviewer #1: Yes

Reviewer #2: Yes

5. Review Comments to the Author

Reviewer #1: Comments in page order

1. Line 44: It's true that threshold elevation after contrast adaptation is substantial, but a bit of an exaggeration to say that "a high contrast grating can cause detection thresholds for similar gratings to increase by orders of magnitude (e.g., Blakemore & Campbell, 1969)." I can't recall any studies finding more than about a 5x increase in threshold, even with high contrast and long duration - ie. less than 1 order of magnitude.

2. Line 91: How were head position and head orientation controlled ? In an experiment on tilt aftereffects, this is important.

3. Was the display gamma-corrected? What was its mean luminance?

4. Line 96-107: Somewhere, need to state both the spatial spread constant (sigma) of the adapting Gabor, and the characteristics (e.g. SF, sigma, duration) of the test Gabor.

5. The illustration of timing in Fig 1B is very confusing: it appears to show the TAE test stimulus being shown immediately after the two secondary task intervals - but the text clearly implies that TAE and seondary stimuli were presented on separate, alternating trials of 1.5s each. Presumably Fig 1B is trying to save space by showing the two kinds of trials in a single sequence - but for the reader this really does not work. It might be clearer if the 4 kinds of trials (secondary vs TAE) x (baseline vs adaptation) were shown separately & clearly labelled, including an indication of when responses were made. The text on its own (p.6) is pretty clear, but Fig 1B creates confusion.

6. Figure 2 is even more mystifying. How the sequence of trials unfolds over time is completely unclear to me. I can't even tell which way the time axis goes (down, up sideways ? Or maybe all three ?). There must be a better way of picturing all this...And better labelling. For example, it is very unclear that the label 'Practice' applies not only to the right-hand column of boxes (close to the label) but also to the left-hand column (rather remote). The authors should try to put themselves in the mind of the reader who is not already highly familiar with the procedure...

7. Line 220 'Figure 2' should be 'Figure 3'

8. Experiment 1: I think there are at least two very significant shortcomings in the results, that relate to both the method used and the way the results are presented.

(i) First, for Fig. 3 (left) it is stated that "Plotted points are estimates of the orientation of the plaid components that appeared square, computed as the average presented orientation on each trial". They are 'estimates', yes, but it seems to me that they are quite likely to be biased ones. With this method of adjustment, the observer is trying to adjust the plaid (over trials) to appear 'square', but during this time the plaid's appearance is changing as the aftereffect decays. Thus there is no way to tell from the data which stimulus presentations might be close to the perceptual null point ('squareness') and which ones are in transition, thus sitting below or above the null point. Perhaps the authors are assuming that these estimation errors will average out, but that seems to me a fairly hazardous assumption, especially when we find out that the effect of interest (a task-dependent difference in the TAE) is really quite small.

(ii) Second, it is stated (L. 227-230) that "To account for across subject differences in baseline (the plaid configuration that appeared square without adaptation), we computed a net TAE by subtracting baseline from adaptation for each observer. " This is a key point, because it is clear from the data (Fig 3) that the baseline the authors subtract is itself task-dependent, and not simply the average baseline over the two tasks. And close inspection of the TAE results (Fig 3) strongly suggests that the main difference in TAE for the two secondary tasks arises from the baseline data, not from the adaptation condition. This aspect of the data seriously undermines the authors' interpretation, that there was "stronger adaptation in the Fixation condition than in the Grating condition. (line 232-3)".

9. Experiment 2: The design of this experiment was tidied up in several ways, compared with Expt 1. The overall result was that there was now no significant effect of task on the TAE (p<0.16, Fig 6). There was, however, a significant effect for the first half of the experiment (Fig. 7). Unfortunately the baseline results for this subset of data are not presented, so we cannot tell what contribution any baseline difference may have made to the effect of task on TAE.

10. Discussion is interesting & wide-ranging, but I think it cannot overcome the basic weakness, or at the very least fragility, of the results. The case that "the processes controlling adaptation are sensitive to the task the observer is performing" is not convincingly supported.

Reviewer #2: The authors examined whether visual adaption (tilt aftereffect) was affected by performance of an additional task. They predicted that “if adaptation depends upon task, then it should be reduced when observers perform the task where adaptation was expected to hurt performance.” They adapted observers to a high contrast grating and measured decay of the tilt aftereffect while they performed a secondary task: either a spatial frequency discrimination on a grating or luminance discrimination on a small dot. In line with their hypothesis, the authors found that that adaptation was smaller when observers performed the spatial frequency discrimination. The conclude that adaptation is reduced when it impairs performance.

Overall, this is a well designed and implemented study. I have one substantive comment and a few minor ones.

1. Throughout the authors assumed that adaptation made performance in the spatial frequency discrimination more “challenging” or “difficult” than the luminance discrimination but provide no data to support this claim. I would encourage them to present discrimination thresholds for both tasks before and after adaptation to substantiate the central tenet of their hypothesis.

Minor:

76: Our sample size was within the range of that used in the prior literature on contrast adaptation

Wouldn’t a statistically power analysis of the required sample size be more appropriate here rather than just doing what other studies have done before?

86: How did they ensure a linear luminance (Gamma) correction on an LCD display (MacBook pro)?

89: typo - “Apparatus”

6. PLOS authors have the option to publish the peer review history of their article (what does this mean?). If published, this will include your full peer review and any attached files.

Reviewer #1: No

Reviewer #2: No

---

## [Author Response · Author response to Decision Letter 0]

9 Jan 2020

Response to reviews: Adaptation Is Weaker When It Harms Task Performance

We thank both reviewers and the editor for their helpful comments. We have addressed each one carefully, and believe that the manuscript is much stronger as a result. Below, we address the concerns point by point.

Reviewer #1: 

We thank the reviewer for the close read and thoughtful response.

Comment: “1. Line 44: It's true that threshold elevation after contrast adaptation is substantial, but a bit of an exaggeration to say that "a high contrast grating can cause detection thresholds for similar gratings to increase by orders of magnitude (e.g., Blakemore & Campbell, 1969)." I can't recall any studies finding more than about a 5x increase in threshold, even with high contrast and long duration - ie. less than 1 order of magnitude.

Response: We thank the reviewer for pointing this out, and have toned down the wording. The revised manuscript now reads (near line 43): 

…exposure to a high contrast grating can cause detection thresholds for similar gratings to more than double.

Comment: “2. Line 91: How were head position and head orientation controlled? In an experiment on tilt aftereffects, this is important.”

Response: Head position and orientation were stabilized with a chin-rest. We apologize for the oversight and now report this in the revised manuscript.

Comment: “3. Was the display gamma-corrected? What was its mean luminance?”

Response: Yes both were done. We now report that (near line 91):

Mean display luminance was 42 candelas/meter2; presented luminances were measured with a PhotoResearch PR-655 and the displayed levels were linearized using software look-up tables.

Comment: “4. Line 96-107: Somewhere, need to state both the spatial spread constant (sigma) of the adapting Gabor, and the characteristics (e.g. SF, sigma, duration) of the test Gabor.”

Response: We thank the reviewer again for noticing our oversight. Sigmas for all Gabors were 1/6 of the stimulus size, and all test durations were 100 msec. We now report these in the revised manuscript.

Comment: “5. The illustration of timing in Fig 1B is very confusing: it appears to show the TAE test stimulus being shown immediately after the two secondary task intervals - but the text clearly implies that TAE and seondary stimuli were presented on separate, alternating trials of 1.5s each. Presumably Fig 1B is trying to save space by showing the two kinds of trials in a single sequence - but for the reader this really does not work. It might be clearer if the 4 kinds of trials (secondary vs TAE) x (baseline vs adaptation) were shown separately & clearly labelled, including an indication of when responses were made. The text on its own (p.6) is pretty clear, but Fig 1B creates confusion.

6. Figure 2 is even more mystifying. How the sequence of trials unfolds over time is completely unclear to me. I can't even tell which way the time axis goes (down, up sideways ? Or maybe all three ?). There must be a better way of picturing all this...And better labelling. For example, it is very unclear that the label 'Practice' applies not only to the right-hand column of boxes (close to the label) but also to the left-hand column (rather remote). The authors should try to put themselves in the mind of the reader who is not already highly familiar with the procedure...”

Response: We completely agree that the old methods figures were confusing, and we have revised them substantially, aiming for simplicity and comprehensibility. Please see revised Figures 1, 2, & 3.

Comment: “7. Line 220 'Figure 2' should be 'Figure 3'”

Response: Corrected.

Comment: “8. Experiment 1: I think there are at least two very significant shortcomings in the results, that relate to both the method used and the way the results are presented.

(i) First, for Fig. 3 (left) it is stated that "Plotted points are estimates of the orientation of the plaid components that appeared square, computed as the average presented orientation on each trial". They are 'estimates', yes, but it seems to me that they are quite likely to be biased ones. With this method of adjustment, the observer is trying to adjust the plaid (over trials) to appear 'square', but during this time the plaid's appearance is changing as the aftereffect decays. Thus there is no way to tell from the data which stimulus presentations might be close to the perceptual null point ('squareness') and which ones are in transition, thus sitting below or above the null point. Perhaps the authors are assuming that these estimation errors will average out, but that seems to me a fairly hazardous assumption, especially when we find out that the effect of interest (a task-dependent difference in the TAE) is really quite small.”

Response: This is point is well taken. To address concerns that averaging the presented orientation is a biased estimate of the point of subjective equality (orientations that produce square checks- neither too tall nor too short) we performed a simulation study. We assumed that simulated observers' true aftereffects decayed exponentially: They had an internal variable that corresponded to their PSE, the orientations that would appear square to them, and this decreased exponentially over time following the rate of our average real observer. For the simulated observers we also used a simple decision rule to determine response, thresholding the difference between the presented stimulus and the value of the PSE variable. Using these rules we simulated 500 observers in our experiment, and found that the running average of presented orientation just slightly overestimated the value of the observers' PSE. We attach a figure showing the results below, and now report the main findings of this simulation in the revised manuscript (near line 181):

TAE results were plotted as estimates of the orientation of the plaid components that appeared square. These were computed as the average presented orientation on each trial. Because the presented orientation was determined by a staircase-like procedure (based on the participants’ responses that attempted to make the checks appear square; see Tasks above), it is possible that simply averaging the presented orientations would produce a biased estimate. To test whether simple averaging of the staircase levels was a reasonable estimate, we performed a Monte-Carlo simulation of our experiment with a model observer responding. The model observer’s orientation that appeared square started at 48 degrees and decayed exponentially over time with a time constant 0.03, which produced a “true” effect similar to those seen in our and other studies. On each simulated trial, noise was added to the presented orientation, and the observer used a simple multiple threshold decision model to pick its response. Averaging across 10,000 simulations, the estimate obtained from simple averaging of the presented orientation fell close to the model observer’s true effect and did not reliably differ from it.

Comment: “(ii) Second, it is stated (L. 227-230) that "To account for across subject differences in baseline (the plaid configuration that appeared square without adaptation), we computed a net TAE by subtracting baseline from adaptation for each observer. " This is a key point, because it is clear from the data (Fig 3) that the baseline the authors subtract is itself task-dependent, and not simply the average baseline over the two tasks. And close inspection of the TAE results (Fig 3) strongly suggests that the main difference in TAE for the two secondary tasks arises from the baseline data, not from the adaptation condition. This aspect of the data seriously undermines the authors' interpretation, that there was ‘stronger adaptation in the Fixation condition than in the Grating condition. (line 232-3)’.”

Response: This is a good point, and the possibility was raised in our original manuscript, and served as the motivation for Exp. 2. We note that while it is possible that the baseline differences can be interpreted differently, there is also an interpretation favorable to our hypothesis: It could be that the lower baseline requires larger adaptation effects to produce the observed lack of difference. We now discuss this more completely in the revised manuscript (near line 286): 

One complication in the data is that performance on the plaid task differed as a function of task during the baseline blocks. Differential effects of attention in the two conditions may explain these differences. When participants attend the gratings in the secondary task display, they may have produced the small amount of contrast adaptation that was measured by the TAE task. This effect may have been reduced in the Fixation Task, when subjects did not attend the gratings. Such an effect, however, should also arise both at baseline and during the adaptation blocks; in this case subtracting the two would cancel it out. It remains possible, however, that the baseline differences were due to unrelated factors, such as noise of some sort, in which case they may explain, all by themselves, the differences in total TAE observed between tasks. Accordingly, we designed Experiment 2 to minimize possible differences in baseline performance.

Comment: “9. Experiment 2: The design of this experiment was tidied up in several ways, compared with Expt 1. The overall result was that there was now no significant effect of task on the TAE (p<0.16, Fig 6). There was, however, a significant effect for the first half of the experiment (Fig. 7). Unfortunately the baseline results for this subset of data are not presented, so we cannot tell what contribution any baseline difference may have made to the effect of task on TAE.”

Response: We now report results more completely for the analysis where they were broken down by day, and show baseline effects for both days in the relevant Figures. While the Exp. 2 day 1 results do contain a small baseline difference, the difference in adaptation is statistically significant even without subtracting the baseline out. This suggests that task indeed does have an effect on adaptation. We report these results in the revised manuscript (near line 333): 

In the first session, total adaptation was reliably weaker when subjects performed the Grating task than when they performed the Fixation task (Figure 8A; p < 0.01, non-parametric signed-rank test used because of non-normal distribution of data). The difference was not reliable in the second testing session (Figure 8B; p > 0.5). In addition, while there was a small baseline difference between tasks in the first testing session, the difference in total adaptation was reliable even without correcting for baseline (p < 0.02, non-parametric signed-rank test).

Comment: “10. Discussion is interesting & wide-ranging, but I think it cannot overcome the basic weakness, or at the very least fragility, of the results. The case that "the processes controlling adaptation are sensitive to the task the observer is performing" is not convincingly supported.”

Response: Hopefully the response to point 9 has allayed the reviewers concerns somewhat. We now acknowledge the possible fragility of effects in the revised manuscript. We believe that they are worth reporting nevertheless, since they make an important, and to our knowledge novel, theoretical point (near line 420):

…the effect of task on adaptation was seen primarily during the first testing session. Our account of why this may be so- that participants learned to perform the secondary task independently of the effects of adaptation- is speculative. It additionally suggests that, in some circumstances, adaptation may not need to be regulated by task. Finally, relative to overall effects of adaptation, its modulation by task was not large in absolute terms. Characterizing when task performance can affect adaptation, and by how much, including in natural viewing during natural tasks, is an important line for future research. Nevertheless, our results provide reasonably strong support that task can affect the strength of adaptation. 

Reviewer #2: 

We thank the reviewer for the helpful comments.

Comment: “1. Throughout the authors assumed that adaptation made performance in the spatial frequency discrimination more “challenging” or “difficult” than the luminance discrimination but provide no data to support this claim. I would encourage them to present discrimination thresholds for both tasks before and after adaptation to substantiate the central tenet of their hypothesis.”

Response: We presented performance on the secondary tasks in Figures 5 and 8 in the original manuscript, but did not discuss these results very extensively. These are new Figures 6 and 9 in the revised manuscript, and we have added discussion of them: 

(near line 269) Finally, we examined whether our assumption, that performance on the Grating task would be more challenged by adaptation than performance on the fixation task, was reflected in the data. Because adaptation was strongest in the first 40 sec of testing, we computed average performance over that interval and the two successive 40 sec periods. Performance on the Grating task was reliably reduced following adaptation during this first interval (Figure 6; t(8) = 2.8, p < 0.03). Performance was reduced numerically, but not reduced reliably for the Fixation task (t(8) = 1.3 p > 0.2), and the difference between the two tasks was not reliable (p> 0.5)….

(near line 342) We again analyzed the performance on the secondary tasks, before and after adaptation (Figure 9). Performance on the Grating task was again reliably reduced following adaptation during the first interval (t(8) = 3.9, p < 0.01). Performance was reduced numerically, but not reliably for the Fixation task (p > 0.5). The difference between the two tasks was not reliable (t(8) = 1.9, p < 0.09).

Comment: “76: ‘Our sample size was within the range of that used in the prior literature on contrast adaptation.’ Wouldn’t a statistically power analysis of the required sample size be more appropriate here rather than just doing what other studies have done before?”

Response: We agree that a power analysis would have been preferable, but because our paradigm was novel, we did not have good a priori estimates of our potential effect size (i.e. the difference in TAE between task conditions) and so we simply used sample sizes from the prior literature generally.

Comment: “86: How did they ensure a linear luminance (Gamma) correction on an LCD display (MacBook pro)?”

Response: We apologize for the oversight in reporting. We measured the display with a spectrophotometer and linearized it in software look-up tables. This has proven stable for LCD displays in our lab in general. We now report this in the revised manuscript: 

Mean display luminance was 42 candelas/meter2; presented luminances were measured with a PhotoResearch PR-655 and the displayed levels were linearized using software look-up tables.

Comment: “89: typo - “Apparatus””

Response: Fixed

---

## [Decision Letter · Decision Letter 1]

21 Jan 2020

PONE-D-19-23461R1

Adaptation is weaker when it harms task performance

PLOS ONE

Dear Dr. Engel,

Thank you for submitting your manuscript to PLOS ONE. After careful consideration, we feel that it has merit but does not fully meet PLOS ONE’s publication criteria as it currently stands. Therefore, we invite you to submit a revised version of the manuscript that addresses the points raised during the review process.

Please consider the recommendations made by Reviewer 1: "Adaptation recovers more quickly when you attend to a low contrast grating than when you don't" (see below). I think this is an interesting alternative explanation of your findings that should be considered in the discussion section of youor manuscript.

We would appreciate receiving your revised manuscript by Mar 06 2020 11:59PM. To enhance the reproducibility of your results, we recommend that if applicable you deposit your laboratory protocols in protocols.io, where a protocol can be assigned its own identifier (DOI) such that it can be cited independently in the future. For instructions see: http://journals.plos.org/plosone/s/submission-guidelines#loc-laboratory-protocols

We look forward to receiving your revised manuscript.

Kind regards,

Mark W. Greenlee

Academic Editor

PLOS ONE

Reviewers' comments:

Reviewer's Responses to Questions

**Comments to the Author**

1. If the authors have adequately addressed your comments raised in a previous round of review and you feel that this manuscript is now acceptable for publication, you may indicate that here to bypass the “Comments to the Author” section, enter your conflict of interest statement in the “Confidential to Editor” section, and submit your "Accept" recommendation.

Reviewer #1: (No Response)

Reviewer #2: All comments have been addressed

2. Is the manuscript technically sound, and do the data support the conclusions?

Reviewer #1: Yes

Reviewer #2: Yes

3. Has the statistical analysis been performed appropriately and rigorously? 

Reviewer #1: Yes

Reviewer #2: Yes

4. Have the authors made all data underlying the findings in their manuscript fully available?

Reviewer #1: Yes

Reviewer #2: Yes

5. Is the manuscript presented in an intelligible fashion and written in standard English?

Reviewer #1: Yes

Reviewer #2: Yes

6. Review Comments to the Author

Reviewer #1: 1. Figs 1-3, showing the stimuli and stimulus sequences are greatly improved; now very much clearer, so that's a major improvement.

2. The presentation of Expt 2, now with baselines illustrated, is also much improved.

3. Many other changes were made, and they all seem fine, and the paper is much clearer as a result.

4. But I do have some comments on the broader interpretation of the results that I think need to be addressed, if only briefly:

In the Discussion, L 392-5, we read: "The adapting stimulus, a high contrast vertical grating patch, was presented under identical neutral attention conditions in all conditions (and [it was presented] prior to task performance). Our experiment measured effects of the secondary task on the decay of adaptation produced under this common condition." This point is clear, and important, but easy to miss. To express it in other words, and rather more specifically, I think your results show that:

performing the grating task caused adaptation to recover *more quickly* than in the fixation task (Fig 8A, right panel). i.e. the observed effect on decay of adaptation was to speed it up.

This implication of the results seems to me important, but not at all salient in the paper. The message that comes through more strongly in the paper, repeated several times in similar words, is this (L.403): "the Grating Task reduced overall adaptation." And similarly in the Abstract: "Tilt-aftereffects were smaller when subjects concurrently performed the grating task than when they performed the fixation task. These results suggest that the control of adaptation is sensitive to task, and that adaptation is reduced when it interferes with performance."

It was becoming quite mysterious to me how the task could influence the level of adaptation induced before the task had even begun. It might perhaps have been an influence from the previous task block on subsequent adaptation. But the idea that the secondary task could influence the recovery process rather than the initial adaptation process seems eminently more plausible, since the task and the recovery co-exist in time. And indeed it has been known for a good while that the recovery time from contrast adaptation is variable, and not a fixed 'time constant'. Greenlee et al (1991) found that recovery time after contrast adaptation increased roughly in proportion to adapting duration, even though the initial level of threshold elevation showed little dependence on adapting duration. The strength of adaptation, and its persistence, were dissociable and depended on adapting contrast, and adapting duration respectively. More recent papers by Engel & colleagues have reported other more complex findings on the timescale of adaptation and recovery.

The present paper wants to conclude (in its title) that "Adaptation is weaker when it harms task performance". But actually there is no evidence (is there?) for a causal connection between the loss of performance in the grating task and the quicker recovery of adaptation that accompanies the grating task. They are correlated but the link might not be causal. Perhaps a more accurate summary one-liner would be: "Adaptation recovers more quickly when you attend to a low contrast grating than when you don't". The authors dismiss a related idea by arguing that extra adaptation caused by attending to the task grating could only increase the level of adaptation, not decrease it (L 395-402).

But consider the following. What causes adaptation to recover quickly or slowly ? There are no detailed models of this, I think, but one general and widely considered idea is that contrast gain controls should accumulate evidence about the range of contrasts in the world and set levels of gain accordingly. Even the salamander's retina does this sort of thing... This raises interesting questions about how long to accumulate the evidence and when to discard old evidence in favour of new. Could it be that the low-contrast test grating provides evidence that the 'current' contrast range is now lower and so requires higher contrast gain than before (ie higher than in the contrast-adapted state). This rise in gain is seen in the data as quicker recovery. [Relatedly, Kwon, Legge et al (2009, JoV) found that exposure to a low contrast world (through a monocular goggle), caused contrast detection to improve and contrast gain to rise, albeit on rather a long timescale.]

This idea that low contrasts might contribute to resetting contrast gain is of course speculative, but seems at least as plausible as the account offered here under the heading of "Adaptation is weaker when it harms task performance".

I'd like to see at least some discussion of these points. I'm happy for the authors to ignore my speculations if they wish, but I think the idea that the loss of grating task performance had little directly to do with the quicker recovery does need addressing. In short, attending to the task grating might itself cause quicker recovery from adaptation.

Reviewer #2: (No Response)

7. PLOS authors have the option to publish the peer review history of their article (what does this mean?). If published, this will include your full peer review and any attached files.

Reviewer #1: No

Reviewer #2: No

---

## [Author Response · Author response to Decision Letter 1]

3 Feb 2020

Reviewer #1: 

We thank the reviewer for the thoughtful and important comments. We have addressed them carefully, and the manuscript has benefitted greatly. 

Comment: "...performing the grating task caused adaptation to recover *more quickly* than in the fixation task (Fig 8A, right panel). i.e. the observed effect on decay of adaptation was to speed it up."

Response: We agree completely, and have revised the manuscript extensively to emphasize this point. We now refer to more rapid decay of adaptation throughout.

Comment: "Perhaps a more accurate summary one-liner would be: "Adaptation recovers more quickly when you attend to a low contrast grating than when you don't". The authors dismiss a related idea by arguing that extra adaptation caused by attending to the task grating could only increase the level of adaptation, not decrease it (L 395-402). But consider the following. What causes adaptation to recover quickly or slowly? There are no detailed models of this, I think, but one general and widely considered idea is that contrast gain controls should accumulate evidence about the range of contrasts in the world and set levels of gain accordingly. Even the salamander's retina does this sort of thing... This raises interesting questions about how long to accumulate the evidence and when to discard old evidence in favour of new. Could it be that the low-contrast test grating provides evidence that the 'current' contrast range is now lower and so requires higher contrast gain than before (ie higher than in the contrast-adapted state). This rise in gain is seen in the data as quicker recovery.

Response: This is an important and interesting theoretical point, and we thank the reviewer for raising it. We have extensively modified the discussion of the paper to address this concern. An overview is: The "standard" model of the interaction between attention and adaptation in the literature is that 1) attention increases gain of neurons in early visual cortex, and 2) this should produce increased response to an attended stimulus, which 3) predicts an increase in adaptation when the adapter is attended. This model is actively debated with some amount of evidence on both sides. The point we want to make in discussion is that this model cannot account for our data, and we think this argument still holds.

The reviewer also makes a different point, which is that attention can modify the >control< of adaptation independently from its effects on the stimulus. To take the reviewer's example, if we attend to low contrast, this fact may be better communicated to whatever neurons control attention, and gain may accordingly be reduced. This is precisely the sort of model we wished to support in our paper, and we apologize that this was not communicated more clearly. We consider attentional control of adaptation a mechanistic account of how task performance can affect adaptation rather than an independent alternative hypothesis. Regarding the title and causality, we prefer to leave task in the title, since after all, task was our independent variable. The relevant part of the discussion now reads: 

More generally, because the visual display was identical in both task conditions, our results also indicate that adaptation is not determined by the stimulus alone. This interpretation agrees with past work showing that changing visual attention can influence the amount of adaptation a given stimulus produces (e.g. Keller et al., 2017; Bartlett, Graf, Hedger, & Adams, 2019). Whether attention can affect adaptation, and by how much, remains a topic of some debate (Bartlett, Graf, Hedger, & Adams, 2019, Morgan and Solomon, 2019). Both sides of this debate assume a similar “standard” model of how attention and adaptation interact: Attention strengthens certain neural responses to the attended stimuli in early visual cortex, and these larger responses in turn produce greater amounts of adaptation. 

This standard model cannot explain our results. The adapting stimulus, a high contrast vertical grating patch, was presented under identical neutral attention conditions in all conditions (and prior to task performance). Our experiment measured effects of the secondary task on the decay of adaptation produced under this common condition. It is theoretically possible, however, that attention influenced some amount of adaptation arising from the secondary task display itself. But such effects are not likely to play a role in our results, since they would be expected to go in the opposite direction from what we observed. In the Grating Task, attention to the grating should boost the neural response it produces, which would be expected to produce adaptation that decreases response to subsequent low-contrast stimuli. Such an effect would be in the same direction as that produced by the high contrast adapting grating, and so should enhance the overall amount of adaptation seen. Importantly, the effect we observed was opposite to this—the Grating Task reduced overall adaptation. 

Attention is, however, a potential mechanism through which task could affect adaptation. In the Grating Task, for example, attention to the low contrast grating could increase the amount of adaptation to that grating. Adaptation to low contrast generally produces the opposite effect of adaptation to high contrast (e.g. Zhang et al., 2009), and so would raise the gain of neurons whose gain responsiveness was reduced by the initial adaptation, causing more rapid decay of the TAE. 

Note that this account proposes a gating effect of attention, on adaptation, which differs from the standard model discussed above. Functionally, this attention-mediated, task-based adaptation to low contrast could serve to center the response to the grating on the steep part of the responding neurons’ contrast response curve (e.g. Barlow et al., 1976). Alternatively, the same adaptation might be characterized as increasing the signal-to-noise of the most important information for the task (e.g. Solari and Serences, 2009). These accounts are not mutually exclusive.

---

## [Editor Report · Decision Letter 2]

5 Feb 2020

Control of visual adaptation depends upon task

PONE-D-19-23461R2

Dear Dr. Engel,

We are pleased to inform you that your manuscript has been judged scientifically suitable for publication and will be formally accepted for publication once it complies with all outstanding technical requirements.

With kind regards,

Mark W. Greenlee

Academic Editor

PLOS ONE
---

## [Editor Report · Acceptance letter]

13 Feb 2020

PONE-D-19-23461R2 

Control of visual adaptation depends upon task 

Dear Dr. Engel:

I am pleased to inform you that your manuscript has been deemed suitable for publication in PLOS ONE. Congratulations! Your manuscript is now with our production department. 

With kind regards,

on behalf of

Dr. Mark W. Greenlee 

Academic Editor

PLOS ONE